# Whole-Genome Resequencing Reveals Signatures of Adaptive Evolution in *Acanthopagrus latus* and *Rhabdosargus sarba*

**DOI:** 10.3390/ani14162339

**Published:** 2024-08-14

**Authors:** Jingyu Yang, Zizi Cai, Yan Fang, Binbin Shan, Ran Zhang, Longshan Lin, Yuan Li, Jing Zhang

**Affiliations:** 1Third Institute of Oceanography, Ministry of Natural Resources, Xiamen 361005, China; yangjingyu@tio.org.cn (J.Y.); caizizi@mails.ccnu.edu.cn (Z.C.); 18110677050@163.com (Y.F.); iamzhangran@163.com (R.Z.); linlsh@tio.org.cn (L.L.); 2School of Life Science, Central China Normal University, Wuhan 430079, China; 3Fisheries College, Jimei University, Xiamen 361021, China; 4South China Sea Fisheries Research Institute, Chinese Academy of Fisheries Sciences, Guangzhou 510300, China; shanbinbin@yeah.net; 5Fujian Provincial Key Laboratory of Marine Fishery Resources and Eco-Environment, Xiamen 361021, China

**Keywords:** *Acanthopagrus latus*, *Rhabdosargus sarba*, whole-genome resequencing, selection signatures, adaptive evolution

## Abstract

**Simple Summary:**

*Acanthopagrus latus* and *Rhabdosargus sarba* are important species for fishing and aquaculture in China’s coastal areas. However, in recent years, their populations have declined due to the impact of marine ecosystem development and aquaculture expansion., and there has been limited research on the adaptation and evolutionary processes of Sparidae facing environmental changes. Therefore, this study used whole-genome resequencing technology to explore the genetic adaptive evolution mechanisms of these species. The results suggest that *Rhabdosargus sarba* may have evolved adaptations related to immunity, feeding, growth, and movement, while *Acanthopagrus latus* may have evolved adaptations related to immunity, nervous system, growth, development, and metabolism. These findings provide important theoretical foundations for the conservation and management of these species.

**Abstract:**

*Acanthopagrus latus* and *Rhabdosargus sarba* are economically important marine species along the coast of China, with similar external morphological characteristics and living habits, with wide distribution and strong adaptability. To investigate the molecular mechanisms underlying the adaptive evolution of these two species, we conducted whole-genome resequencing of 10 individuals of both species from the coastal waters of Wuyu Island, Fujian, China, using high-throughput sequencing technology. We obtained SNP, InDel, CNV, and SV variation information and annotated these variations, constructing a genomic variation database for both species. By comparing the resequencing data with reference genomes, we identified 9,829,511 SNP loci in the population of *A. latus* and 34,051,056 SNP loci in the population of *R. sarba*. Using whole-genome SNP data, we employed Fst and ROD methods to identify candidate genomic regions under selection. Functional annotation and enrichment analysis using GO and KEGG databases revealed potential adaptive evolution in *R. sarba* associated with immune response, feeding, growth and development, and locomotion, while *A. latus* showed potential adaptive evolution associated with immune response, nervous system, growth and development, and metabolism.

## 1. Introduction

Yellowfin seabream (*Acanthopagrus latus*) and goldlined seabream *(Rhabdosargus sarba*) are both members of the Actinopterygii, Perciformes, Sparidae family. These species hold significant economic value in both capture fisheries and aquaculture industries along the Chinese coast. They share similar ecological niches, primarily inhabiting tropical and subtropical coastal waters, with minimal long-distance migrations. Their distribution spans across the East China Sea, South China Sea, southern Japan, Philippine waters, and the warm waters of the Indian–Western Pacific region [1,2]. Despite their morphological similarities, several distinguishing features allow for the differentiation between these species; for example, the yellowfin seabream exhibits a yellow coloration on the lower portion of its caudal fin, contrasting with the dark hue and black margins observed in goldlined seabream. Additionally, differences in snout sharpness and lateral stripe patterns are also important features to distinguish the two species. However, they exhibit significant differences in their life history characteristics and reproductive strategies. The goldlined seabream is a rudimentary hermaphrodite with no pronounced size-based gender differences, whereas the yellowfin seabream is a protandrous hermaphrodite, with smaller individuals, predominantly male, and larger individuals, predominantly female, optimizing egg production. The yellowfin seabream typically shows a wide size distribution within populations, whereas the goldlined seabream displays a more uniform size range among captured individuals. The protandrous nature of the yellowfin seabream enhances reproductive success by allowing larger females to produce more eggs while smaller males effectively fertilize them. Behaviorally, the yellowfin seabream does not exhibit size-based grouping during spawning, reflecting diverse size distributions, whereas the goldlined seabream tends to form groups of similar-sized individuals, indicating different social dynamics [3]. In recent years, rapid coastal development, intensified marine aquaculture practices, and escalating fishing activities have imposed significant pressures on marine ecological systems. These anthropogenic impacts have led to declines in coastal marine biodiversity and resource abundance, consequently affecting the populations of both yellowfin seabream and goldlined seabream, to varying extents [4].

Currently, research on goldlined seabream and yellowfin seabream mainly focuses on biology (such as reproductive [5] and growth characteristics [6,7]), physiological ecology [8,9], and genetics). Among these, genetic studies primarily involve phylogeny and genetic structure [10,11]. For instance, Liu et al. [12] employed the mitochondrial DNA control region to assess genetic diversity and population structure in yellowfin seabream populations from three sites in Dongshan Bay and one in Xiamen, China, thereby evaluating their genetic resource status. Jiang et al. [13] conducted a molecular systematics study of four sparid fish species, including yellowfin seabream and goldlined seabream, based on mitochondrial cytochrome b sequences. This research elucidated the classification and phylogenetic relationships of sparid fish at the molecular level. However, owing to the maternal inheritance of mitochondrial genes, they are limited in their capacity to reflect paternal genetic information [14,15]. This limitation underscores the necessity for a dependable molecular-marker technology that encapsulates a broader spectrum of genetic information to effectively address this issue.

Whole-genome resequencing (WGR) technology is a method of comprehensively sequencing individuals or populations within a species at the genome level. This approach facilitates the detection of high-density genetic variation sites across the genome, thereby offering a thorough understanding of the genomic information of the studied species [16,17]. Therefore, WGR serves as a powerful tool in population genomics research, primarily utilized to investigate issues pertaining to population genetic structure, genome evolution, and the conservation of genetic diversity in evolutionary biology [18,19,20,21].

Currently, there are few genetic studies on the Sparidae employing whole-genome sequencing, and there are no studies examining the adaptive evolution of yellowfin seabream and goldlined seabream utilizing WGR. Based on the whole-genome resequencing technology, this study investigated the adaptive evolution of yellowfin seabream and goldlined seabream in the vicinity waters of Wuyu Island, and detected the genetic variation information in both species, establishing a comprehensive genomic variation database. Subsequently, we used genomic data to elucidate the mechanisms driving adaptive evolution in these two Sparidae species. The findings of this study bridge existing knowledge gaps regarding adaptive evolution in Sparidae fishes, thereby providing a scientific foundation for evolutionary research within this family. Moreover, our insights pave the way for further investigations into adaptive evolution among species exhibiting sympatric distributions.

## 2. Materials and Methods

### 2.1. Ethics Statement

All animal procedures were conducted in accordance with the guidelines and regulations set forth by the Experimental Animal Management and Ethics Committee of the Third Institute of Oceanography, Ministry of Natural Resources (Xiamen, China), and appropriate approvals were obtained prior to commencement of the study.

### 2.2. Source of the Samples

The yellowfin seabream (WA, *n* = 10) and goldlined seabream (WY, *n* = 10) samples utilized in this study were obtained from the waters surrounding Wuyu Island, Fujian Province, China (Figure 1), during October 2022. These samples were obtained from the same location and fishing batch of individually hand-caught specimens. To ensure sample representativeness and comparability, random-sampling methods were employed. Healthy individuals were selected from a large catch, excluding those with evident external injuries, diseases, or parasitic infections. The yellowfin seabream specimens had a length range of 178–190 mm and a weight range of 176.3–217.9 g, while goldlined seabream specimens ranged in length from 164 to 189 mm and weighed between 159.3 and 208.9 g. All samples were collected from dorsal muscle tissue, fixed in anhydrous ethanol, and stored at −80 °C for subsequent experiments; fresh dorsal muscle tissue was directly used for whole-genome DNA extraction.

### 2.3. Sequence Data and Variant Detection

Marine animal genomic DNA extraction was conducted using a TIANamp Marine Animals DNA Kit (TIANGEN BIOTECH (BEIJING) Co., Ltd., Beijing, China). To verify the quality of the extracted DNA, we used the NanoDrop 2000 spectrophotometer (Thermo Fisher Scientific Inc., Waltham, MA, USA) to determine the concentration of DNA samples and assess their purity. The integrity of genomic DNA was evaluated through agarose gel electrophoresis. The specific sample-submission standards were the following: DNA concentration ≥ 100 ng/μL, total amount ≥ 50 μg, and A260/A280 ratio between 1.8 and 2.0. Electrophoresis showed no significant RNA bands, and the genomic bands were clear and intact, with the main band expected to be above 100 kb. Subsequently, DNA library preparation and sequencing were performed by Personalbio technology Co., Ltd. (Shanghai, China).

Both yellowfin seabream and goldlined seabream populations were aligned to the reference genome of a mixed assembly of the yellowfin seabream genome (*A. latus* fAcaLat1.1.fa, NCBI BioProject database under Bioproject ID PRJEB40700), which comprised second and third generations. Variant detection was executed using reads with a mapping quality (map Q) of ≥20 and properly paired alignments. Sequences near insertions and deletions (InDels) were realigned and recalibrated using GATK (v4.2.0) software to enhance the accuracy of SNP positions around InDels [22]. GenotypeGVCFs from the GATK (v4.2.0) toolkit were employed for variant detection analysis to acquire precise genotypes. The identified SNPs and InDels underwent filtering using the VariantFiltration module, and the specific filtering parameters are as follows: for SNPs, use QD < 2.0 || MQ < 40.0 || FS > 60.0 || SOR > 3.0 || MQRankSum < −12.5 || ReadPosRankSum < −8.0; for InDels, use QD < 2.0 || FS > 200.0 || SOR > 10.0 || MQRankSum < −12.5 || ReadPosRankSum < −8.0. And the resulting VCF files containing high-quality SNPs and InDels were annotated using the Annovar software (https://annovar.openbioinformatics.org/en/latest/, (accessed on 30 July 2024)) [23]. Furthermore, structural variations (SVs) and copy number variations (CNVs) were identified using lumpy (v0.2.13) and CNVnator software (v0.4.1) [24], respectively. Subsequently, their detection results were annotated using the Annovar software.

### 2.4. PCA Analysis

Principal component analysis (PCA) is a method that clusters individuals into distinct subgroups based on genomic SNP differences among all individuals within a population. In this study, the whole-genome SNP dataset, obtained through screening, was formatted using vcftools (v0.1.16) and plink (v2.0) software. Subsequently, PCA plots were generated using GCTA (v1.94) software, utilizing various principal component features to cluster individuals based on their eigenvalues.

### 2.5. Genetic Diversity Analysis and Candidate Genes for Selection

Using vcftools software (v0.1.16) [25], the populations were partitioned into 10 kb steps with 100 kb windows for inter-population calculations, resulting in fixation index (Fst) and overall population nucleotide diversity (π_θ_) values for each population. The reduction in diversity (ROD) was computed based on π_θ_ values, where ROD = |1 − π_θ_/π_θXM_|. The results of Fst and ROD were integrated, and the top 5% of Fst and ROD values were selected to identify regions of the genome under selection. Subsequently, selective-sweep analysis was conducted on the populations of yellowfin seabream and goldlined seabream from Wuyu Island, leading to the identification of candidate genes.

### 2.6. Functional Annotation and Enrichment Analysis of Candidate Genes

The selected candidate genes underwent functional annotation using the GO and KEGG databases [26,27,28,29,30]. Gene Ontology analysis entailed GO term enrichment of candidate genes within the selected regions, followed by the identification of candidate genes significantly associated with biological functions. Similarly, KEGG pathway analysis was conducted on candidate genes within the selected regions to gain insights into their biological functions.

## 3. Results

### 3.1. Whole-Genome Resequencing Data of Yellowfin-Seabream and Goldlined-Seabream Populations

Whole-genome resequencing was conducted on 10 individuals each of yellowfin seabream and goldlined seabream from the waters around Wuyu Island. For yellowfin seabream, deep resequencing yielded a total of 88.13 Gb raw data, averaging 8.81 Gb per sample, with an average sequencing depth of 12.72×, a mean GC content of 42.76%, and average Q20 and Q30 scores of 97.29% and 92.91%, respectively. Post-filtering, approximately 85.90 Gb of clean data were obtained. Subsequent variant detection and correction identified 9,829,511 SNP loci, 3,394,141 InDel loci, 34,239 SV loci, and 2819 CNV loci within the yellowfin seabream population.

For goldlined seabream, the raw data amounted to approximately 85.55 Gb, with an average of 8.56 Gb per sample, an average sequencing depth of 12.36×, an average GC content of 42.67%, and average Q20 and Q30 scores of 97.09% and 92.57%, respectively. After filtering, approximately 83.01 Gb of clean data were obtained. Variant detection and correction identified 34,051,056 SNP loci, 9,510,492 InDel loci, 37,287 SV loci, and 4937 CNV loci in the goldlined seabream population.

### 3.2. Genetic Population Structure of the Yellowfin-Seabream and Goldline-Seabream Individuals

Principal component analysis (PCA) was conducted on single-nucleotide polymorphisms (SNPs) derived from samples of yellowfin seabream and goldlined seabream collected from the waters of Wuyu Island. Group distinctions were represented by different colors on the PCA scatter plot. In Figure 2, the PCA plot (PC1, PC2 and PC3) shows the genetic structure of the 10 yellowfin seabream individuals (WA) and 10 goldlined seabream individuals (WY). The degrees of explained variance is given in parentheses, and the PCA plot illustrated clear differentiation between individuals of yellowfin seabream and goldlined seabream across three dimensions: PC1 (contribution rate: 11.92%), PC2 (contribution rate: 1.49%), and PC3 (contribution rate: 1.35%). And the points of yellowfin seabream are clustered together, while those of goldlined seabream are more dispersed.

Moreover, population nucleotide diversity (π_θ_) was computed based on SNP data for both yellowfin seabream and goldlined seabream. The analysis revealed higher nucleotide diversity in yellowfin seabream (π_θwa_ = 3.246 × 10^−3^) compared to goldlined seabream (π_θwy_ = 1.381 × 10^−3^).

### 3.3. Detection of Genes Related to Adaptive Evolution in Yellowfin-Seabream and Goldlined-Seabream Populations

We identified selective regions within the genomes of yellowfin seabream and goldlined seabream, yielding 574 candidate genes. Functional enrichment analysis was conducted on these selected regions, and the candidate genes underwent Gene Ontology (GO) and Kyoto Encyclopedia of Genes and Genomes (KEGG) enrichment analyses. The GO analysis (Figure 3a) revealed that these candidate genes were predominantly enriched in processes such as glial cell development, myelination, regulation of glial cell differentiation, glycogen metabolic process, and the cellular-glucan metabolic process. KEGG analysis (Figure 3b) indicated enrichment of these candidate genes in pathways including acute myeloid leukemia, colorectal cancer, prostate cancer, endometrial cancer, thyroid cancer, and basal-cell carcinoma, etc.

In yellowfin seabream and goldlined seabream sampled from Wuyu Island, the selected genes included *NCOR2*, *FOXO3B*, *TCF7L2*, *NF1A*, *CAPN*, *FTH1B*, *TBC1D32*, *SESN1*, *CLEC16A*, *ALDH1L1*, *LRRC75A*, *EP400*, *PIDD1*, *PCDH15A*, and *CLEC11A*, etc. (Figure 4 and Figure 5).

## 4. Discussion

Yellowfin seabream and goldlined seabream share similar ecological habits as those of demersal marine fishes, primarily dwelling in shallow coastal waters and exhibiting limited long-distance migrations. Both species are omnivorous, with a broad dietary spectrum that including algae, mollusks, crustaceans, and organic debris [7,31].

This study compares the genetic diversity, population structure, and selection signals between yellowfin seabream and goldlined seabream populations (designated as WA and WY) from Wuyu Island, China. We focused on genomic variations, to elucidate their adaptive evolution mechanisms in this region. The selected genes in these genomes may be closely associated with their ecological habits and habitat, providing them with advantages for survival and reproduction in specific environmental conditions. These findings not only deepen our understanding of the genetic adaptation of these two fish species, but also offer important insights for future conservation and resource management efforts.

This study, based on PCA results, clearly distinguishes between the two study subjects, yellowfin seabream and goldlined seabream, where the yellowfin seabream exhibits a more dispersed distribution of points in PCA analysis compared to the clustered distribution seen in the goldlined seabream. This observation suggests that yellowfin seabream may possess higher genetic diversity at the genomic level. Furthermore, comparisons of nucleotide diversity reveal that the yellowfin seabream displays greater nucleotide polymorphism, a trait often associated with stronger adaptive capabilities to environmental changes, implying a richer genetic resource under environmental pressures and fluctuations. These findings align with those of Hesp, S.A. [3], where similar conclusions were drawn regarding the biological characteristics of yellowfin seabream and goldlined seabream: in terms of reproductive strategies, the goldlined seabream employs a dispersed spawning strategy, enhancing the survival rates of some eggs and juveniles while reducing predation risks. In contrast, the yellowfin seabream adopts a concentrated spawning strategy, despite its higher risks, allowing for the production of more eggs and juveniles within a shorter timeframe, exceeding the feeding capacity of predators. Additionally, the yellowfin seabream’s longer lifespan mitigates the adverse effects of annual recruitment variability on population size. Overall, the yellowfin seabream demonstrates higher reproductive success and a more diversified adaptive strategy, reflecting its complex ecological niche. Conversely, the goldlined seabream tends towards a more conservative reproductive strategy, favoring the formation of stable population structures.

In addition, our study identifies a series of genes such as *NCOR2*, *FOXO3B*, and *TCF7L2*, which may be involved in adaptive evolution in yellowfin seabream and goldlined seabream populations from Wuyu Island.

The *NCOR2* gene encodes the NCOR2 protein, also known as the T3 receptor-associated co-repressor 1 (TRAC-1) or silencing mediator of retinoic acid and thyroid hormone receptor (SMRT). *NCOR2* participates in transcription factor interactions during B-cell development, and is a significant member of the thyroid hormone and retinoic acid receptor-associated co-repressor family [32,33]. Ding et al. [32] identified specific expression of the *NCOR2* gene in hematopoietic stem cells and secondary hematopoietic sites in zebrafish. Knockdown or knockout of the *NCOR2* gene results in a reduction in the number of T cells and hematopoietic stem cells. In this study, we observed strong positive selection on the *NCOR2* gene in goldlined seabream. Therefore, compared to yellowfin seabream, this observation may suggest that the *NCOR2* gene plays a crucial role in hematopoietic stem cell formation, B-cell, and T-cell development during the adaptive evolution process of goldlined seabream.

The *FOXO* gene regulates various cellular processes including cell proliferation, oxidative-stress response, reproduction, apoptosis, and longevity, and it is implicated in vascular homeostasis and embryonic development [34,35]. Studies have indicated high expression of the *FOXO3B* gene in ovarian germ cells of *Epinephelus coioides* and *Gobiocypris rarus*, suggesting its involvement in folliculogenesis or oogenesis in fish [36,37]. In this study, we observed strong positive selection on the *FOXO3B* gene in goldlined seabream. Therefore, compared to yellowfin seabream, the *FOXO3B* gene may play a crucial role in embryonic development, reproduction, cell proliferation, and apoptosis processes in goldlined seabream.

*TCF7L2* is a transcription factor of significant importance in the Wnt/β-catenin signaling pathway, which plays a critical regulatory role in developmental processes. Previous studies have demonstrated that TCF7L2 protein expression gradually increases during the induction of differentiation in 3T3L1 and primary adipocyte precursor cells. Additionally, TCF7L2 protein expression is necessary for Wnt signaling to regulate adipocyte generation [38], and it plays an important role in adipose tissue development and function. In this study, we observed strong positive selection on the *TCF7L2* gene in yellowfin seabream. Therefore, compared to goldlined seabream, the *TCF7L2* gene significantly influences the growth and development of adipocytes in yellowfin seabream.

*NF1A* belongs to the nuclear factor 1 family, acting as a site-specific DNA-binding protein. In the cerebral cortex, the *NF1A* gene participates in astrocyte genesis and serves as a downstream target gene in the Notch signaling pathway. *NF1A* prevents excessive astrocyte generation, primarily by directly inhibiting the transcription of the *HES1* target gene, forming a negative-feedback regulation loop [39]. In this study, we observed strong positive selection on the *NF1A* gene in yellowfin seabream. Combined with the results of GO enrichment analysis, genes were found to be enriched in processes such as glial cell development, regulation of glial cell differentiation, and oligodendrocyte differentiation. This suggests that, compared to goldlined seabream, the *NF1A* gene plays a significant role in the formation, development, and differentiation of glial cells in yellowfin seabream.

In yellowfin seabream, we identified the *CAPN1*, *CAPN2L*, *CAPN8*, and *CAPN2B* genes. *CAPN1* gene is a Ca^2+^ regulated protease that plays a role in cell death, apoptosis, and cellular motility [40]. The calpain family comprises proteolytic enzymes crucial for affecting meat tenderness by participating in the hydrolysis of muscle proteins. *CAPN* genes are pivotal in influencing meat tenderness, significantly impacting protein metabolism and growth processes in fish [41]. Compared to the goldlined seabream, *CAPN* genes play an important role in muscle growth, development, and protein metabolism in the yellowfin seabream, contributing to its superior flesh quality over goldlined seabream.

Additionally, this study identified genes such as *FTH1B*, *CLEC11A*, and *CLEC16A* undergoing strong positive selection in goldlined seabream. The *FTH1B* gene is involved in the development of pharyngeal teeth in fish, and plays a role in early mineralization of teeth [42]. In this study, yellowfin seabream and goldlined seabream were sampled from the same habitat and are both omnivorous fish. However, they may differ in food selection and nutrient sources, which could influence their requirements for minerals and other nutrients, thereby affecting the expression and function of the FTH1B gene. If the goldlined seabream relies more on a diet containing higher mineral content, its FTH1B gene may undergo stronger positive selection. *CLEC11A* is involved in bone-development regulation, while *CLEC16A* is associated with various immune disorders, playing a crucial role in immune development [43]. On the other hand, genes such as *TMEM*, *DDB1*, *VPS13C*, *TBC1D15*, *NLRC5*, *BSX*, *RTN4A*, and *LAMA1* were found to undergo strong positive selection in yellowfin seabream. The *TMEM* gene plays a crucial role in immune processes, participating in multiple physiological processes such as the activation of signaling pathways, adhesion, cell chemotaxis, autophagy, and apoptosis [44]. *DDB1* is involved in regulating adipose tissue development and physiological processes related to obesity [45]. The *VPS13C* gene is associated with high insulin levels [46], while *TBC1D15* plays a key role in regulating Rab GTPase activity and sugar absorption [47]. The *NLRC5* gene is an important regulatory gene in adaptive and innate immune responses, controlling type I interferon, nuclear factors-κB, the expression of MHC class I gene, and inflammation signaling pathways [48]. *BSX* is a key molecule controlling pineal-gland gene expression and development, which is crucial for pineal neuron development and differentiation. The pineal gland is a brain organ that secretes the sleep hormone melatonin and perceives light in many animals [49]. *RTN4A* is a marker of oligodendrocytes, and can inhibit the growth of nerve dendrites [50]. *LAMA1* participates in protein synthesis during muscle production and development processes [51].

Therefore, in the study of adaptive evolution of yellowfin seabream and goldlined seabream based on whole-genome resequencing in the same habitat, the following findings were observed: (1) Immune Processes: genes *NCOR2* and *CLEC16A* are associated with the immune processes of goldlined seabream, while in yellowfin seabream, *TMEM* and *NLRC5* genes are linked to immune processes; (2) Growth, Development, and Metabolic Processes: genes *FTH1B*, *CLEC11A*, and *FOXO3B* are associated with feeding, growth, development, and locomotion in goldlined seabream. However, in yellowfin seabream, genes *TCF7L2* and *DDB1* are related to its fattening process, while genes *NF1A*, *RTN4A*, and *BSX* are associated with its nervous system, with the *BSX* gene enhancing its light-sensing ability. Additionally, genes *CAPN* and *LAMA1* promote muscle growth, development, and protein metabolism, while genes *VPS13C* and *TBC1D15* facilitate the metabolism of yellowfin seabream.

## 5. Conclusions

In summary, this study used whole-genome resequencing technology to obtain SNP, InDel, CNV, and SV variation information of yellowfin seabream and goldlined seabream from Wuyu Island, and conducted detailed annotations, constructing their genomic variation databases. High-quality SNP analysis revealed that the yellowfin seabream exhibits higher genetic diversity and nucleotide polymorphism at the genomic level compared to the goldlined seabream, suggesting a stronger adaptive capacity to environmental changes and more complex conditions. Furthermore, through the selection of candidate genes, this study elucidated adaptive evolution within the same habitat for both species: goldlined seabream potentially evolved adaptations related to immunity, feeding, growth and development, and locomotion, while yellowfin seabream may have adapted in terms of immunity, nervous system, growth and development, and metabolism. The research results provide a scientific basis for the evolutionary study of Sparidae fish, and serve as a reference for studies on adaptive evolution among co-distributed species.

## Figures and Tables

**Figure 1 animals-14-02339-f001:**
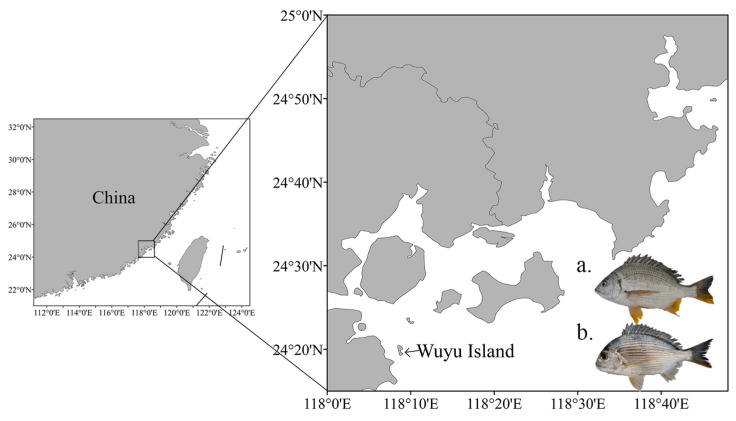
The geographic map of sample sources for yellowfin seabream (**a**) and goldlined seabream (**b**) in this study.

**Figure 2 animals-14-02339-f002:**
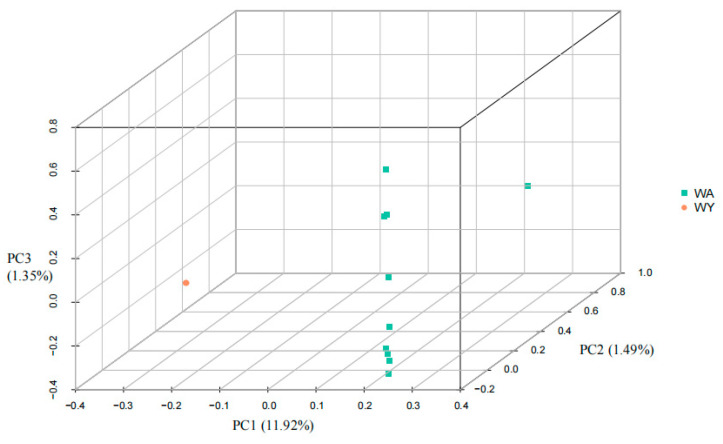
PCA plot (PC1, PC2 and PC3) showing the genetic structure of the 10 yellowfin seabream individuals (WA) and 10 goldlined seabream individuals (WY). The degrees of explained variance are given in parentheses.

**Figure 3 animals-14-02339-f003:**
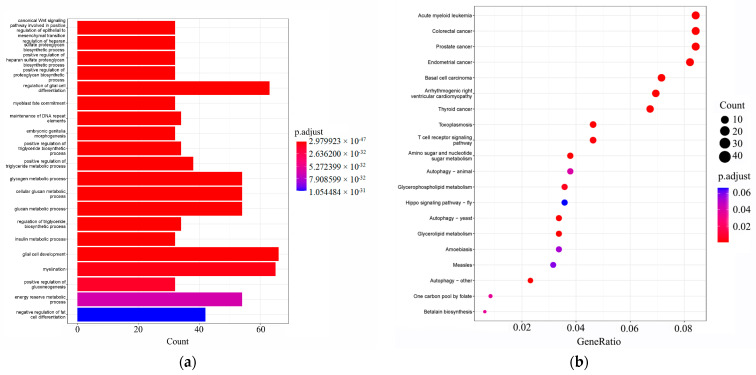
Enrichment analysis of GO (**a**) and KEGG (**b**) of selected genes.

**Figure 4 animals-14-02339-f004:**
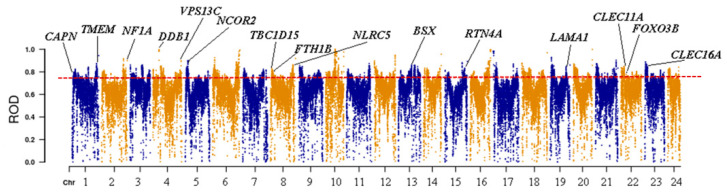
The Manhattan plot of the ROD values along chromosomes. The yellow/blue dots represent the ROD value of all SNPs, and the red dashed line represents the threshold line of the top 5% of ROD.

**Figure 5 animals-14-02339-f005:**
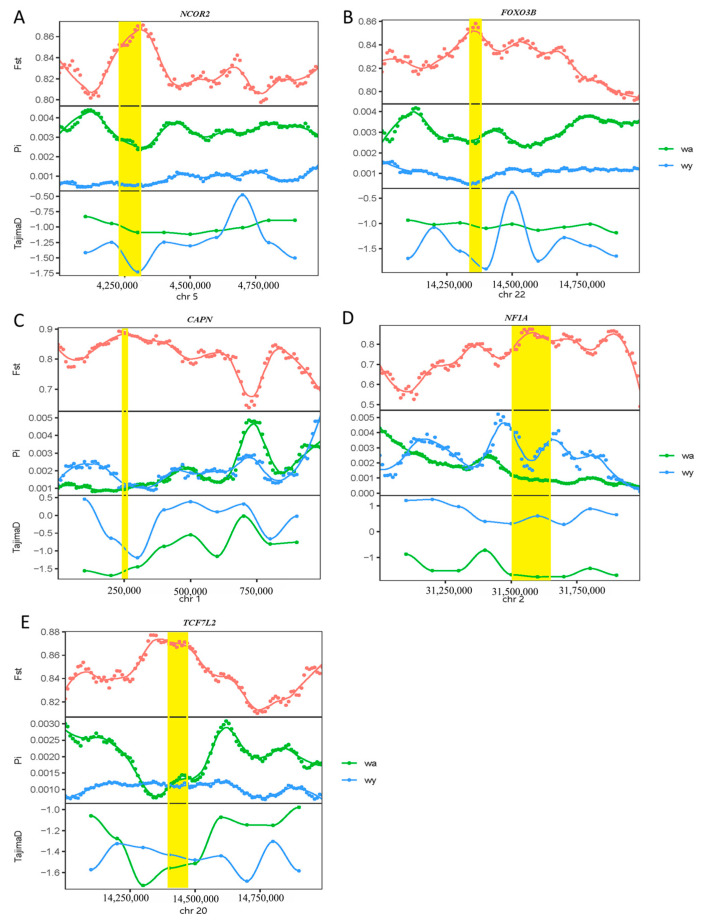
(**A**) The value of F_ST_, π_θ_ and Tajima’s *D* near gene *NCOR2*. (**B**) The value of F_ST_, π_θ_ and Tajima’s *D* near gene *FOXO3B*. (**C**) The value of F_ST_, π_θ_ and Tajima’s *D* near gene *CAPN*. (**D**) The value of F_ST_, π_θ_ and Tajima’s *D* near gene *NF1A*. (**E**) The value of F_ST_, π_θ_ and Tajima’s *D* near gene *TCF7L2*. The yellow highlight indicates gene regions with strong selective signals.

## Data Availability

The datasets used in this study are available in online repositories. The study data have been deposited in the NCBI repository under the BioProject accession number PRJNA1141615.

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
