# Peer review of "Whole-Genome Resequencing Reveals Signatures of Adaptive Evolution in Acanthopagrus latus and Rhabdosargus sarba"

_animals, 2024, doi:10.3390/ani14162339_

Round 1

Reviewer 1 Report

Comments and Suggestions for Authors

The present research «Whole Genome Resequencing Reveals signatures of Adaptive Evolution in Acanthopagrus latus and Rhabdosargus sarba» provides an interesting and relevant work on the genetics and adaptive evolution of two fish subspecies. The authors provided a comprehensive report on the sequencing results of both species groups, each consisting of 10 individuals. The present study was performed at a high scientific level, thus, the obtained results are informative. This manuscript can be accepted for publication after refining the following points:

  1. It is advisable to provide phenotypic characteristics of the studied fish specimens (including weight, body length, size and other available features).
  2. Line107. Authors should provide information on the supplier of the marine animal genomic DNA isolation kit (manufacturer and country).
  3. I suggest to make the conclusion less general, and instead list the specific results.
  4. Line73. I think the indefinite article «an» is a misprint and should be replaced with «of».
  5. Line79. there is a few of genetic research => there are a few genetic studies
  6. Line80. Have explored => examining
  7. Line85. we employ => we used
  8. Line100,185,188,194-195,. The reference is missing
  9. Line168. Showing => shows
  10. Figure 3(b) the legend is partially missing
  11. Different candidate genes associated with adaptive selection have been identified for different groups of traits in the two fish subspecies. Perhaps the authors could make some assumptions about the origin and purpose of fixing these differences. For example, is it possible that FTH1B in goldlined seabream underwent strong positive selection due to the emergence of a nutritional advantage compared to yellowfin seabream?

Reviewer 2 Report

Comments and Suggestions for Authors

The paper entitled “Whole Genome Resequencing Reveals signatures of Adaptive 2 Evolution in Acanthopagrus latus and Rhabdosargus sarba” is a comparative genomic/population study of two seabream fish species from coast of China.

Although the study has great potential, there are important methodological doubts that prevent validation of the results and conclusions presented by the authors.

Abstract section:

1.       In lines 32 to 35, the project design is not clear. is it a population analysis? or a comparative genomic analysis?

Important: In this context, the title, the objective and the proposed methodology must be congruent with each other; therefore, it is recommended to check that they are consistent with each other.

Introduction section

the introduction section should be improved in the following issues:

1.       In addition to the morphological comparison of both species, differences in behavior, productivity and specific niche, or ecological aspects such as interspecific competition between these two species (for example, do the two species exploit the same trophic niche?) that justify a possible pattern of adaptive convergence or not, should be included. it is also recommended to postulate a clear hypothesis in this regard.

2.       Previous research on morphometrics, molecular biology, genetics and genomics of both species should be expanded. The description presented by the authors is weak. These previous results could improve the justification of the study. In this case, for example, how would this work differ from previous research such as Xi et al., 2008 (https://doi.org/10.1111/j.1095-8649.2008.02010.x) showing genetic structure in Acanthopagrus latus?

Materials and Methods section:

In this section there are important uncertainties in different methodological aspects that the authors should clarify in order to validate the results.

1.       More Important: In Source of the samples section (lines 97 to 102), the process of obtaining the sample (n: 10 of each species) should be expanded.

a.       What were the sampling criteria (inclusion and exclusion) of these 10 individuals?

b.       Were these 10 individuals sampled at different points to ensure greater variability? Or are they all from the same school?

c.       Was it a random or convenience sampling?

d.       MORE IMPORTANT: Could such a low number of 10 individuals (possibly family members; see previous comment) validate the conclusions proposed for a whole species? How do the authors justify it?

These points are important to validate the results presented.

2.       In the line 100, the bibliographic source (Error! Reference source not found.) , must be verified

3.       In lines 107 to 108, in the marine animal genomic DNA extraction kit should include the company of and whether the company's conditions were used for the DNA extraction process. Likewise, how was the extracted DNA verified?

4.       MORE IMPORTAN: the authors used the A. latus fA-111 caLat1.1.fa reference genome for Acanthopagrus latus. This is correct, but it is not clear why they used the same reference genome to identify polymorphisms in Rhabdosargus sarba when there are different genomic sources for this species in the SRA database (for example, SRX10917579). Why wasn't an assembly made from this data?

Reviewer 3 Report

Comments and Suggestions for Authors

The authors present a manuscript entitled "Whole Genome Resequencing Reveals signatures of Adaptive Evolution in Acanthopagrus latus and Rhabdosargus sarba", The work seems to be properly conducted and scientifically sound. I do have a few questions to be addressed:

Several occurrences of "Error! Reference source not found."

Line 168, "Figure 2." should be Figure 2.

Figure 3, is partially cut.

What were the criteria for filtering with VariantFiltration?

How was performed the alignments, e.g. tools used?

Relative to PCA results and nucleotide diversity comparisons, how can these genetic differences confer advantages or disadvantages in their natural habitats? This should be included in the discussion.

It's important to share all the data, embargoed or not.

Round 2

Reviewer 2 Report

Comments and Suggestions for Authors

Although most of the comments were correctly addressed and included in the manuscript, Comment 4 on Sourse of the sample section (now lines 112 to 120) is still not answered correctly. the authors included the morphological characteristics of the individuals collected (lines 115 to 118), but did not answer important details of the sampling.

The following points have not been satisfactorily clarified and included in the manuscript:

a)      the process of obtaining the sample (n: 10 of each species) should be expanded.

b)      What were the sampling criteria (inclusion and exclusion) of these 10 individuals?

c)       Were these 10 individuals sampled at different points to ensure greater variability? Or are they all from the same school?

d)     Was it a random or convenience sampling?

e)      MORE IMPORTANT: Could such a low number of 10 individuals (possibly family members; see previous comment) validate the conclusions proposed for a whole species? How do the authors justify it?

Since this study has a population-based impact, these aspects should be clarified in the document in order to validate its results.
